# Onset Timing of Hyoid Muscles Activation during Cervical Flexion Is Position-Dependent: An EMG Study

**DOI:** 10.3390/life12070949

**Published:** 2022-06-24

**Authors:** Hirofumi Sageshima, Dagmar Pavlů, Dominika Dvořáčková, David Pánek

**Affiliations:** Faculty of Physical Education and Sport, Charles University, 162 52 Prague, Czech Republic; sageshima.hirofumi@ftvs.cuni.cz (H.S.); dvorackova.dominika@ftvs.cuni.cz (D.D.); panek@ftvs.cuni.cz (D.P.)

**Keywords:** surface electromyography (EMG), cervical flexion, suprahyoid muscles, infrahyoid muscles

## Abstract

Due to the high prevalence of neck pain, considerable attention is paid to the function of cervical flexor muscles. Although the deep and superficial cervical flexor muscles have been evaluated, the impact of hyoid muscles on cervical flexion is still not well known. We, therefore, aimed to investigate the activation of hyoid muscles during physiological cervical flexion, and to determine the impact of different starting positions on cervical flexion muscle activation. The activities of bilateral sternocleidomastoid, scalene, suprahyoid, and infrahyoid muscles were evaluated by surface electromyography (EMG) in twenty young healthy volunteers. They performed a repetitive cervical flexion-extension movement, from neutral position of the head to the maximum flexion with the same speed set at eight seconds in a cycle, in three various positions (sitting, standing, and supine). In sitting and standing positions, the group of suprahyoid muscles was activated in advance of other cervical flexor muscles despite only significant differences were found in scalene muscles, and the activation of the group of infrahyoid muscles was time-synchronous with sternocleidomastoid muscles. On the other hand, in supine position, the activation of all measured cervical flexor muscles was significantly earlier and longer than in the other two positions. This study confirmed the empirical suggestion that hyoid muscles contributed to cervical flexion, and it confirmed that muscle activation was position dependent, even if the given movement is nearly identical.

## 1. Introduction

In recent decades, cervical disorders and related neck pain became a common medical problem and one of the leading causes of disability, with up to 71% of lifetime prevalence all over the world [1,2]. The causes of neck pain are variable, but it is considered that the lifestyle-related causes, including computer/smartphone usage and sedentary jobs, are risk factors for nontraumatic neck pain [3,4,5]. Furthermore, due to long-term computer usage, the population of young adults with nonspecific neck pain is increasing [6,7], despite old age being reportedly one of the risk factors for this problem [8].

Due to the increased incidence of neck pain in population, an enormous amount of studies have been published to understand the possible causes of this issue and its impact on the cervical regions in kinematic and electromyographic ways. From these previous findings, deep cervical flexor muscles (longus capitis, longus colli, and rectus capitis anterior muscles) showed lower strength and endurance under neck pain conditions [9]; meanwhile, superficial cervical flexor muscles (sternocleidomastoid and scalene muscles) showed enhanced activation as a compensation [10,11]. This compensatory activation of superficial cervical flexor muscles can be caused by pain or any other pathological conditions leading to a change in muscle activity [12,13], and it was found in patients with cervicogenic headaches [14] and whiplash-related disorders (WAD) [15].

One of the most common methods used for muscle activity evaluation is electromyography (EMG), and its findings delivered valuable information to understand the impact of neck pain on the cervical muscles. In EMG studies, significant attention was given to the evaluation of superficial cervical flexor muscles, mostly sternocleidomastoid and scalene muscles [10,16,17,18]. On the contrary, to date, only a small number of studies evaluated the role of hyoid muscles during cervical flexion [19,20,21,22], and, to the best of our knowledge, no study reported the timing and the relationship of activation of these muscles with other superficial cervical flexor muscles. Hyoid muscles have mainly been investigated in swallowing studies, as these muscles attach to the hyoid bone and the contraction of these muscles pulls the hyoid bone upward and forward during deglutition [23]. However, hyoid muscles contributed to the cervical flexion movement by generating flexion moments in the whole cervical spine, due to their location placed on anterior regions in the cervical spine [21,22]. The latest findings showed that sternohyoid muscle was working as a synergist of cervical flexion combined with sternocleidomastoid muscle [21,24]. These findings were also supported by a study investigating the effect of hyoid muscles on moment-generating capacity and dynamic simulations via a musculoskeletal computer model [25]. However, the activation of hyoid muscles was previously evaluated only during isometric tasks in the sitting position [21], and the craniocervical flexion test (CCFT) in the supine position [17,26]. Even though the chief complaint of pain and discomfort from patients with neck disorders is frequently during physiological neck movements, such as flexion, lateral bending, and rotation [18], only a few studies evaluated the activation of the superficial cervical muscles during physiological cervical movement [18,27,28,29].

From the empirical and practical knowledge and experience, muscle activation is also dependent on extrinsic factors and could be influenced by starting positions, even if the given movement in different starting positions is kinematically identical. As pain and pathologic conditions affect muscle activation [12,13], the gravitational loads also change the way of muscle activity [30]. In a previous study that evaluated trunk muscle activity, the differences in onset time and pattern of muscle activation were found during the cyclic trunk flexion movement from standing and supine positions due to the gravitational loads [30]. It showed that the trunk flexor muscles had higher activation during the deepest flexion in the standing position, while these muscles activated from the beginning to the end of the trunk flexion–extension motion, except around the deeper flexion. However, in the cervical muscles, the influence of different starting positions on the activation of these muscles has not been clarified yet.

The present surface EMG study had two aims. The primary aim was to detect the activation of hyoid muscles along with the other cervical flexor muscles during physiological cervical movements. Based on the present findings [22], we hypothesized that hyoid muscles activate similarly to the sternocleidomastoid and scalene muscles as synergists of cervical flexion. The second aim was to determine the impact of different starting positions on the cervical flexor muscle activation, and we assumed that the supine position increases the cervical flexor activation more than the other positions.

## 2. Materials and Methods

### 2.1. Participants

Twenty voluntary healthy university students (10 females and 10 males) aged 23.3 ± 2.70 were involved in the experiment, and their physical characteristics are shown in Table 1. The study was conducted in the period from July 2021 to September 2021. The exclusion criteria were acute/chronic neck pain, and histories of neck injury, orthopedic disorders, or surgery. The range of motion in cervical spine was measured in all of the participants before EMG measurement, and we confirmed that all of them had physiological range of motion. The study was approved by Charles University FTVS Ethics Committee (EK 079/2020, 19 February 2020). Written informed consent was obtained from each participant after explaining the study protocol.

### 2.2. Methods

The experiment was conducted using a telemetric data transmission technique with the sampling frequency of 1500 Hz, with Noraxon TeleMyo 2400 transmitter (Noraxon U.S.A. Inc., Scottsdale, AZ, USA). Surface electrodes were disposable, self-adhesive Ag/AgCl snap electrodes for surface EMG applications only, designed for both research and clinical use with hypo-allergenic gel and adhesive. The dimension of the figure of eight shaped adhesive was 40 mm × 22 mm; the diameter of the two circular adhesives was 10 mm, and the inter-electrode distance was 20 mm.

The EMG data were collected from paired group of suprahyoid (digastric, mylohyoid, and geniohyoid muscles), group of infrahyoid (omohyoid, sternohyoid, sternothyroid, and thyrohyoid muscles), sternocleidomastoid, and scalene muscles. Locations of electrodes on suprahyoid muscles were the midway between the inferior top of the mandible and the thyroid cartilage by Falla et al. [26]. For infrahyoid muscles, electrodes were positioned midway between the superior attachment on the body of hyoid bone and the inferior attachment on the manubrium and the clavicle bone, and their location was chosen to minimize the crosstalk from sternocleidomastoid muscle by voluntary cervical lateral bending to the ipsilateral side and cervical rotation to the contralateral side by O’Leary et al. [17]. Electrodes on sternocleidomastoid and scalene muscles followed the recommendation by Falla et al. [31]. For sternocleidomastoid muscle, the electrodes were placed on the 1/3 mark on the line from the sternal notch and mastoid process. The electrodes on scalene muscles were along the lateral border of the clavicular portion of sternocleidomastoid muscle. Before applying surface EMG electrodes, the skin was prepared by shaving and cleaning used textile with alcohol to enhance the adherence of the electrodes [32]. Figure 1 shows all of the surface electrode placements.

All participants performed cyclic cervical flexion–extension from neutral to maximum flexion, and vice versa in three different positions—sitting, standing, and supine—in random order. In the cervical flexion phase, the procedures used by Kendall et al. and Page et al. [12,33] were adopted. So, participants initially tucked the chin to flatten the cervical lordosis and then bent the head anteriorly to the maximum range of motion. Each cycle lasted 8 s (4 s from neutral to maximum cervical flexion, and vice versa) and these movements were acoustically controlled by a metronome with 60 bpm [21] to minimize the influence of velocity on the motion itself and EMG signals [34]. Participants repeated this movement 15 times, and we chose 6 cycles so that at least 6 repetitions were sufficient for the analysis of muscle activation in movement patterns [35]. Among the measurements of EMG activity in each starting position, a two-minute rest was included to eliminate the effect of fatigue [36]. Data were collected by trained physiotherapists under the supervision of medical doctors specialized in neurosciences.

### 2.3. Data Proceeding

All of the collected data (raw EMG signals) were processed by the computer program biomechanical analysis software MR 3.8.30 (Noraxon U.S.A. Inc., Scottsdale, AZ, USA) following the Noraxon manuals [32].

As for the determination of the cycle, we segmented and divided it into the flexion phase (from neutral to maximum flexion) and the extension (from terminal to neutral) manually based on the footage shot during measurements. Thereafter, 8 consecutive cycles were selected out of 15, except the first and final two cycles. In the processing of EMG signals, firstly the band-pass filter was applied at 10–500 Hz [32]. Then, the signals from electrocardiography (ECG) were reduced with the ECG reduction tool, as this biological artifact is inevitable in the research for neck [37,38], upper trunk, and upper extremities muscles [32]. After rectification, signals were smoothed by root-mean-square (RMS) with a 100 ms window to produce the linear envelope. Each channel was normalized with the highest value recorded in the same channel in each participant. The EMG signals of 8 consecutive cycles were averaged and normalized with respect to time, and the duration of the cervical flexion–extension cycle was described as a percentage ranging from 0 to 100% (starting position; 0%, maximum cervical flexion; 50%). We calculated the mean RMS value in all of the measured muscles among three positions. Additionally, the onset and offset of muscle activation were calculated with the standard deviation (SD) range of the electromyographic baseline which multiplication of value was defined at three [32], and it calculated the average onset and offset time during 8 consecutive cycles.

### 2.4. Data Analysis

All of the statistical data were sorted out by statistical software, SPSS Statistics 28 (IBM SPSS Inc., Chicago, IL, USA). The onset and the duration time of the muscle activation were compared with Kruskal–Wallis test among muscles of interest in each position. In addition, the onset and duration time of muscle activation, and the means of the EMG value were compared among three positions in four paired groups of muscles using Friedman test. Setting a confidence interval percentage of 95%, and *p* < 0.05 was considered as evidence of showing a significant difference.

## 3. Results

All of the participants performed cyclic cervical flexion in the three different positions, and a variety of muscle activations were acquired from the four paired groups of muscles (suprahyoid, infrahyoid, sternocleidomastoid, and scalene). Figure 2 shows the representative raw EMG signals from all measured muscles among three positions.

Figure 3 depicts the timing of EMG during a cycle of cervical flexion–extension movement in paired four muscles—suprahyoid, infrahyoid, sternocleidomastoid, and scalene—among sitting (a), standing (b), and supine (c) positions. In sitting position, suprahyoid muscles began their activation in advance of sternocleidomastoid and infrahyoid muscles which were followed by scalene muscles; however, no significant difference was found. Likewise, the onset of muscle activation while standing showed a similar tendency to sitting, and significant differences were only found between right suprahyoid muscles and bilateral scalene muscles (*p* < 0.05). In addition, the duration time of activation in the left suprahyoid was significantly longer than in the left scalene muscles (*p* < 0.05). On the other hand, in supine position, every muscle of interest was coordinatingly activated from the very beginning of the cycle. The activation of bilateral scalene muscles was significantly longer than infrahyoid muscles (*p* < 0.05). In addition, significantly earlier onset and longer duration in every muscle were found in supine than in sitting and standing positions (*p* < 0.01).

The means of root-mean-square values of EMG are depicted in Figure 4, compared among three positions in suprahyoid, infrahyoid, sternocleidomastoid, and scalene muscles. In each measured muscle, the value in supine was significantly larger than sitting and standing (*p* < 0.01).

## 4. Discussion

In this study, we aimed to detect the activation of hyoid muscles during physiological cervical flexion and to determine the impact of different starting positions on cervical flexion muscle activation. During cervical flexion movement in both sitting and standing positions, suprahyoid muscles were activated earlier than the other cervical flexor muscles. Then, the activation of sternocleidomastoid and infrahyoid muscles followed. The scalene muscles became involved in the movement only after the involvement of the previous muscles. In the sitting position, there was no significant difference observed in the onset and the duration of muscle activity among any measured muscles. However, in the standing position, suprahyoid muscles were activated significantly earlier and longer than scalene muscles. From the findings of the supine position, all measured cervical flexor muscles were activated from the very beginning of cervical flexion movement. The scalene muscles were activated significantly longer than infrahyoid muscles in the supine position. In the comparison among positions, the onset of measured muscles in supine position was significantly earlier than in the other two positions. Likewise, the duration of muscle activity in the supine position was significantly longer in all of the measured muscles compared to the other two positions. Additionally, the value of EMG amplitude in the supine position was significantly higher than in sitting and standing positions.

Suprahyoid muscles started activating past 20% in advance of the other flexor muscles in sitting and standing positions. This group of muscles, composed of digastric, stylohyoid, mylohyoid, and geniohyoid muscles, attaches from the mandible to the hyoid bone, and their functions are widely known for elevating and drawing hyoid bone forward during swallowing [39,40]. In addition, a previous study claimed that suprahyoid muscles are activated from 5 to 20 degrees of cervical flexion and their activation is increased by the angle inclination [19]. Mortensen et al. [25] reported that the upper cervical spine was stabilized by suprahyoid muscles to resist the extension moment produced by sternocleidomastoid in the simulation of the computer model. On the other hand, infrahyoid muscles commenced the activation around 30% in concert with sternocleidomastoid muscle following suprahyoid muscles. The main factor of infrahyoid muscles, composed of omohyoid, sternohyoid, sternothyroid, and thyrohyoid muscles, a link between hyoid bone and sternum and clavicular bone, is depressing the hyoid bone during swallowing. However, infrahyoid muscles also contribute to cervical flexion. Due to the anterior location of the cervical spine, this group of muscles adds more flexion moments and synchronously contracts along with sternocleidomastoid during voluntary cervical flexion [21,25]. With the contraction of both suprahyoid and infrahyoid muscles, the flexion movement was produced in the head on the cervical spine and in the cervical spine on the thoracic spine when the simultaneous contraction of mastication muscles fixed the mandible [41]. In addition to that, in the view of the kinematics of the hyoid bone, to which both suprahyoid and infrahyoid muscles attach, the movement of this bone correlated to the head, jaw, and upper cervical spines during flexion and extension [42].

As for other flexor muscles, sternocleidomastoid muscle was activated following suprahyoid muscles and coordinating with infrahyoid muscles during cervical flexion in sitting and standing positions. Vasavada et al. [43] reported that sternocleidomastoid muscle produced significant cervical flexion torque on the mid-to-low cervical spine, and the flexion moment-generating capacity increased in the deep cervical flexion. For the scalene muscles, their activation was following other flexor muscles and significantly delayed onset than suprahyoid muscles. Scalene muscles, especially anterior scalene, were activated as cervical flexors and contributed to the stability of the middle to lower cervical spine while contracting bilaterally, despite a limited moment arm to bent cervical flexion ventrally [44]. The activation of these superficial muscles is necessary to complete the deep flexion by overcoming the resistance that the posterior tissues generate [45].

However, in the supine position, the ways of all measured muscles activation were totally different from the other two positions. The activation of suprahyoid muscles led other muscles in sitting and standing positions, whereas each muscle started activation from the very beginning of a given cervical movement in supine. In addition, there were significantly higher means of EMG amplitudes in all of the measured muscles in supine than in sitting and standing positions (Figure 4). Since cervical flexion movement from supine position was against the gravitational force, it was considered higher loads than in the other two positions. Apart from the gravitational loads, cervical flexor muscles needed to overcome the resistance from the dorsal soft tissues, such as the nuchal ligament, to complete the given movement [45]. With these elements, which cervical flexor muscles needed to be against to reach the maximum cervical flexion, cervical flexor muscles performed higher activation and more coordination from the beginning. This result was supported by a previous EMG study, which indicated that a higher load task had earlier onset, later offset, and/or longer muscle activation than a lower load task. In addition, the value of EMG activity in higher load tasks was significantly higher than in lower load tasks [46].

Furthermore, regarding the duration time of muscle activation, it was significantly longer in all of the muscles in supine position than in the others. All of the cervical flexor muscles activated from the beginning to over 70% of the cycle, but this result was different from previous findings. In a study comparing the duration time of trunk flexor muscles between supine and standing positions, it is indicated that the activations of trunk flexor muscles, rectus abdominis and obliquus externus abdominis, were not detected around deeper flexion phases in which the activation of trunk extensor muscles as antagonists was taken over [30]. An explanation would be the difference in the range of motion between the trunk and cervical flexion; the former is beyond 90 degrees, while the latter maximumly reaches around 60 degrees [47]. It is considered that an external extension moment was imposed on the head in a whole cycle, therefore the activations of cervical flexor muscles were recorded across nearly the whole cycle.

A limited number of studies reported hyoid muscle activation during cervical flexion movement. However, one report claimed that some individuals would compensate with hyoid muscles [11]. Thus, the present findings would help to understand the contribution of hyoid muscles as cervical flexor muscles to physiological cervical flexion and the individual strategy of cervical flexion that is often affected by pain and disorders.

One of the limitations of this present study was the technical feature of surface EMG. The conventional surface electrodes were used for evaluating the cervical muscles. Even though it was technically unable to evaluate the activation from cervical deep layer muscles, their activation was examined with innovative surface electrodes or fine-wire [24,48]. Thus, this study was not able to find out the timing of activation and coordination between hyoid muscles and deep muscles during physiological cervical flexion, even though deep muscles were the first muscles to activate and had a shorter reaction time to contribute to the spinal stability [49,50]. Furthermore, participant selection was another limitation of this study. All of them were young healthy adults without any problems in cervical regions, so the impact of pain on the activation of cervical flexor muscles, including hyoid muscles, during physiological cervical flexion is still unknown.

## 5. Conclusions

The present study confirmed the empirical assumption that suprahyoid and infrahyoid muscles are activated differently during physiological cervical flexion. Suprahyoid muscles were activated in advance of the other flexor muscles, while infrahyoid muscles were activated in concert with sternocleidomastoid muscle in sitting and standing positions. On the contrary, in supine, all of the measured cervical flexor muscles had significantly earlier activation and their activation lasted significantly longer than in the other positions, since in this position cervical flexor muscles had to overcome the gravity load. In addition, it also confirmed that positional differences influenced the way of muscles activation, even if a given movement is nearly identical.

## Figures and Tables

**Figure 1 life-12-00949-f001:**
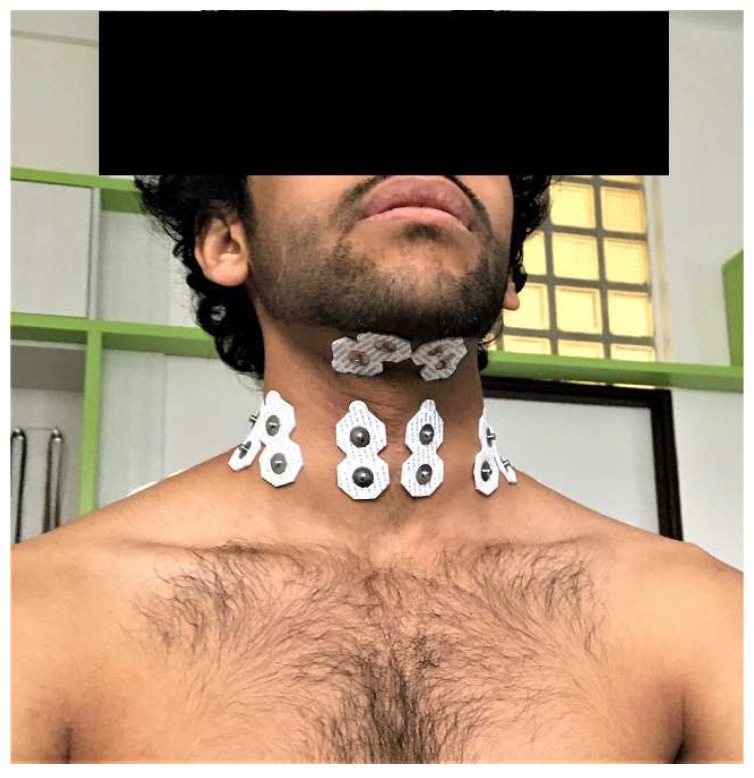
The placement of surface electrodes.

**Figure 2 life-12-00949-f002:**
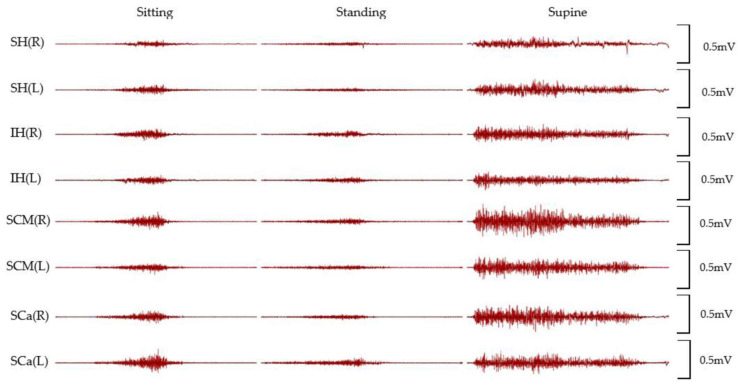
Raw EMG signals of 4-paired muscles in different positions from a representative participant. SH: the group of suprahyoid muscles, IH: the group of infrahyoid muscles, SCM: sternocleidomastoid muscle, Sca: scalene muscles.

**Figure 3 life-12-00949-f003:**
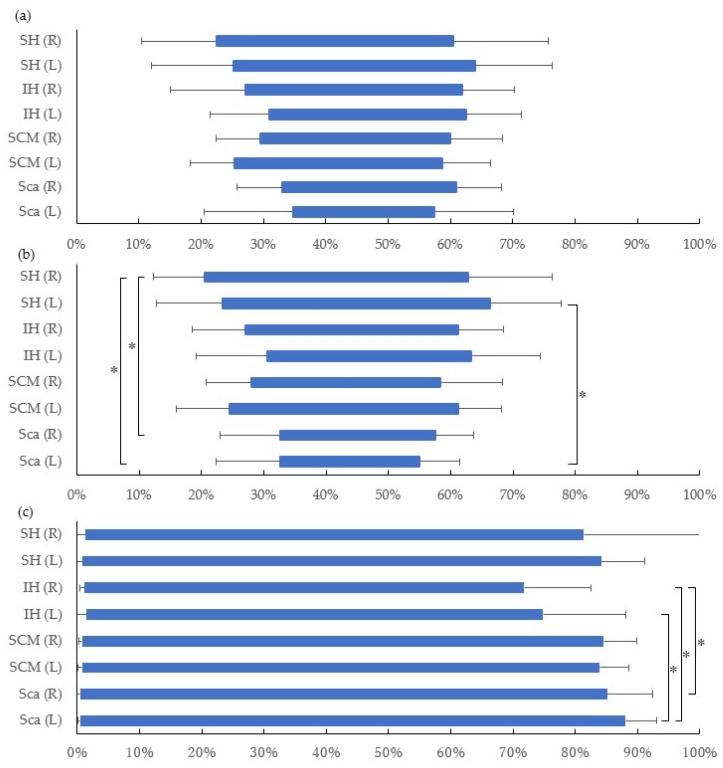
The mean duration of EMG activity from 4-paired muscles was depicted by bars in sitting position (**a**), in standing position (**b**), and in supine position (**c**) as a percentage of one cervical flexion–extension cycle. In standing position, right suprahyoid muscles activated significantly earlier than bilateral scalene muscles (*p* < 0.05), while left suprahyoid muscles activated significantly longer than left scalene muscles (*p* < 0.05). In supine position, all of muscles activated significantly earlier and lasted significantly longer than in the other positions (*p* < 0.01). The activation of bilateral scalene muscles was significantly longer than infrahyoid muscles (*p* < 0.05). Additionally, significantly earlier onset and longer duration in every muscle were found in supine than in sitting and standing positions. SH: the group of suprahyoid muscles, IH: the group of infrahyoid muscles, SCM: sternocleidomastoid muscle, Sca: scalene muscles. *: *p* < 0.05.

**Figure 4 life-12-00949-f004:**
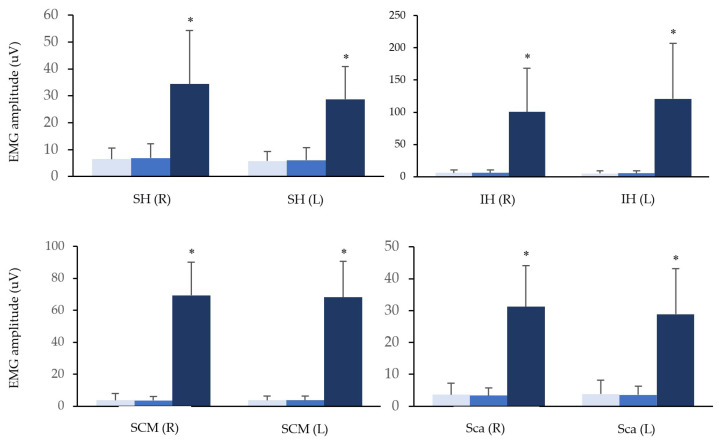
Data for the mean of EMG amplitude in a cycle of cervical flexion among different positions depicted on 4-paired muscles. Sitting (light blue), standing (blue), supine (dark blue). SH: the group of suprahyoid, IH: the group of infrahyoid, SCM: sternocleidomastoid, Sca: scalene. *: *p* < 0.01.

**Table 1 life-12-00949-t001:** Basic characteristics of participants.

	Age	Height (m)	Weight (kg)	BMI
Participants (*n* = 20)	23.3 (2.70)	1.70 (0.08)	66.8 (12.4)	23.0 (3.22)

## Data Availability

The datasets generated for this study are available on request to the corresponding author.

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
