# Peer review of "Onset Timing of Hyoid Muscles Activation during Cervical Flexion Is Position-Dependent: An EMG Study"

_life, 2022, doi:10.3390/life12070949_

Round 1

Reviewer 1 Report

Thank you for allowing me to review the revisions to this manuscript.

The authors have addressed my concerns. I am pleased to inform you that this revised manuscript is now suitable for publication.

Reviewer 2 Report

I have read them and my position is to recommend the acceptance of the manuscript in its current state. I believe the authors have successfully responded to my comments and that the manuscript is now suitable for publication.

This manuscript is a resubmission of an earlier submission. The following is a list of the peer review reports and author responses from that submission.

Round 1

Reviewer 1 Report

I appreciate the opportunity to review this manuscript. However, I am quite concerned about the experimental methodology. I am not convinced that surface emg is sufficient to detect the activation of these very small muscles. Because of their small size, I am not convinced they are not detecting crosstalk from all the closely packed muscles. Figure 1 supports this. Further, the muscle patterns and the representative activation do not match. This is concerning. These small muscles would require indwelling electrodes to be accurate. To make this paper appropriate for publication, I would need to see a photograph of electrode placement and some methodology that demonstrates that crosstalk was not detected.

Next, I suggest some editing from a native English speaker. 

Author Response

REW: I appreciate the opportunity to review this manuscript. However, I am quite concerned about the experimental methodology. I am not convinced that surface emg is sufficient to detect the activation of these very small muscles. Because of their small size, I am not convinced they are not detecting crosstalk from all the closely packed muscles. Figure 1 supports this. Further, the muscle patterns and the representative activation do not match. This is concerning. These small muscles would require indwelling electrodes to be accurate. To make this paper appropriate for publication, I would need to see a photograph of electrode placement and some methodology that demonstrates that crosstalk was not detected.

Next, I suggest some editing from a native English speaker.

AUTHORS: Dear reviewer, thank you for taking the time to read our manuscript. In your evaluation, you raise some doubts about surface electromyography and the evaluated muscles. In this context, we would like to state that in our study, individual supra- and infrahyoid muscles were not evaluated, but always as a group. When applying the surface electrodes, we proceeded in the same way as in the studies that we cite in the text. We have also added a new image to the text (Fig. 1), which demonstrates the application of electrodes. In the edited text, all the changes we have made are marked in red. These are fixes that have been recommended by all reviewers. The text was also corrected by a native speaker.

Reviewer 2 Report

The manuscript “Electromyographic analysis of hyoid muscles during cervical flexion movement” aimed to investigate the activation of the hyoid muscles during cervical flexion starting from different positions. I commend the author for their work in conceptualizing the experiment, collecting the data and discussing the obtained results. General comments and specific points and sections are provided below:

General comments

Presentation: English writing should be revied by a professional. Overall, the text is well written, but there are many language issues that could be easily fixed by a professional reviewer. I commend the authors for their excellent work in presenting data in figures.  

Title: the title is adequate and adheres to the scope of the paper. However, I suggest that the authors include their main finding in the title to increase the discoverability and research outreach in case the manuscript is accepted. Bear in mind that this is only a recommendation that the authors should feel free to accept or reject.

Abstract: the abstract should provide more information to readers. Please clarify the methods and present actual results within the abstract. This reviewer was not able to understand the experiment when reading the abstract.

Introduction: as a whole, the introduction provides the rationale for the study. However, the three last paragraphs of the introduction should be revised and perhaps reordered. The way the authors present the aims and hypotheses are a little confusing. Please try to provide the entire rationale of the study before presenting the aims and hypotheses. The final paragraph is especially confusing and most of it should be introduced prior to the aims, within the introduction.

Materials and Methods: sample size and characteristics are adequate to address the research question.  Methods are well described and controlled. The analyses are also clearly described and are in line with the state of the art on EMG assessment and analysis. Statistical procedures are adequate.

Results: I would recommend that the authors conjugate the verbs in the present for figures, instead of past (i.e., “are depicted” instead of “were depicted”).  Tables and figures are well-designed and provide a clear overview of the obtained data.

Discussion: the discussion is brief and concise, as should be. The authors succeed in providing an overview of the obtained results while contextualizing them with the available literature. The content of the discussion is rich in kinesiological aspects, and I commend the authors for their interesting interpretations. Limitations are acknowledged, especially the one that was my main concern: the use of surface EMG. The one thing missing in the discussion, in my opinion, was an applied perspective for the obtained data. Not all experiments should be directed to practical applications, but they should be pointed out when relevant.

Conclusion: the conclusion is short and concise as should be.

Please find specific comments detailed below:

L16-17: Please rephrase “they performed with a 15-fold (…) various positions” for clarity

L39: to date?

L40: increased incidence?

L57-59: Please rephrase “Generally, these muscles are (…) the deglutition [24].” For clarity

L130: the procedures adopted by sound better than the “way from”

L185: depicts?

Author Response

R: The manuscript “Electromyographic analysis of hyoid muscles during cervical flexion movement” aimed to investigate the activation of the hyoid muscles during cervical flexion starting from different positions. I commend the author for their work in conceptualizing the experiment, collecting the data and discussing the obtained results. General comments and specific points and sections are provided below:

General comments

Presentation: English writing should be revied by a professional. Overall, the text is well written, but there are many language issues that could be easily fixed by a professional reviewer. I commend the authors for their excellent work in presenting data in figures. 

AUTHORS: Dear Reviewer, thank you for taking the time to read our manuscript and for the notes you have written to us.  We comment on all your comments and suggestions below, and in the text we have marked all the changes we have made in red. The text was corrected by a native speaker

R: Title: the title is adequate and adheres to the scope of the paper. However, I suggest that the authors include their main finding in the title to increase the discoverability and research outreach in case the manuscript is accepted. Bear in mind that this is only a recommendation that the authors should feel free to accept or reject.

AUTHORS: Thank you for recommending that we change / edit the manuscript title. We thought a lot and would like to present a changed headline that perhaps more reflects "what was done in our study"

R: Abstract: the abstract should provide more information to readers. Please clarify the methods and present actual results within the abstract. This reviewer was not able to understand the experiment when reading the abstract.

AUTHORS: We have fundamentally reworked the abstract according to your recommendation, all changes are marked in red.

R: Introduction: as a whole, the introduction provides the rationale for the study. However, the three last paragraphs of the introduction should be revised and perhaps reordered. The way the authors present the aims and hypotheses are a little confusing. Please try to provide the entire rationale of the study before presenting the aims and hypotheses. The final paragraph is especially confusing and most of it should be introduced prior to the aims, within the introduction.

AUTHORS: We fully agree with you, we have restructured and supplemented the Introduction section.

R: Materials and Methods: sample size and characteristics are adequate to address the research question.  Methods are well described and controlled. The analyses are also clearly described and are in line with the state of the art on EMG assessment and analysis. Statistical procedures are adequate.

AUTHORS: Thank you for the positive evaluation of the procedures we used.

R: Results: I would recommend that the authors conjugate the verbs in the present for figures, instead of past (i.e., “are depicted” instead of “were depicted”).  Tables and figures are well-designed and provide a clear overview of the obtained data.

AUTHORS: Thank you for your comments and recommendations, all discrepancies have been corrected

R: Discussion: the discussion is brief and concise, as should be. The authors succeed in providing an overview of the obtained results while contextualizing them with the available literature. The content of the discussion is rich in kinesiological aspects, and I commend the authors for their interesting interpretations. Limitations are acknowledged, especially the one that was my main concern: the use of surface EMG. The one thing missing in the discussion, in my opinion, was an applied perspective for the obtained data. Not all experiments should be directed to practical applications, but they should be pointed out when relevant.

AUTHORS: Our study is the initial study in a given topic, so our results may become the basis for further studies. In addition, our results can help to understand the issue. In addition to the chapter Discussion, we added the following text: "Thus, the present findings would help the understanding of the contribution of hyoid muscles as cervical flexor muscles to physiological cervical flexion and the individual strategy of cervical flexion that is often affected by pain and disorders"

R: Conclusion: the conclusion is short and concise as should be.

AUTHORS: Thank you for the positive evaluation of the procedures we used.

R: Please find specific comments detailed below:

L16-17: Please rephrase “they performed with a 15-fold (…) various positions” for clarity

L39: to date?

L40: increased incidence?

L57-59: Please rephrase “Generally, these muscles are (…) the deglutition [24].” For clarity

L130: the procedures adopted by sound better than the “way from”

L185: depicts?

AUTHORS: Thank you for your comments on specific passages, all inconsistencies have been corrected in the text

Reviewer 3 Report

Thank you for submitting this paper to Life. The manuscript under consideration: "Electromyographic analysis of hyoid muscles during cervical flexion movement" is an interesting article on an important topic in Life. However, there are a few major concerns.

1. "The number of people suffering from this condition could have been significantly increased during the COVID-19 pandemic, as computer usage became more common in the general population. However, no evidence has supported this assumption in short term effect the date [9]." This sentence is not relevant to the current study. We recommend that it be removed.

2. Consider whether the period of confinement by the COVID pandemic (covered by the study) has had an effect on both the modification of some of the parameters assessed. 

3. Please clarify the duration of the survey. Does the survey period include the COVID-19 epidemic period?

4. Consider adding data on range of motion of the cervical joints.

5. Who performed the tests? How was the data stored ? Consider a flow diagram regarding recruitment process and data collection.

6. I think you are using the wrong statistical method. Since the data are from the same participants, the data must be reanalyzed to a corresponding test. The statistical analysis that should be used in this study is the Friedman test. The method of multiple comparisons is then commonly used.

7. Please clarify how the different positions affect and help in daily life, and how the different activation of the suprahyoid and infrahy-oid muscles during physiological cervical flexion may affect and help in daily life.

Author Response

REW: Thank you for submitting this paper to Life. The manuscript under consideration: "Electromyographic analysis of hyoid muscles during cervical flexion movement" is an interesting article on an important topic in Life. However, there are a few major concerns.

AUTHORS: Dear reviewer, thank you very much for the time you gave to our manuscript, and also thank you very much for all the recommendations you made. We have been careful to incorporate all suggestions and we believe that any adjustments have contributed to raising the level of our article. We comment on each of your comments below, and we have made all the corrections in red to the manuscript.

REW:

  1. "The number of people suffering from this condition could have been significantly increased during the COVID-19 pandemic, as computer usage became more common in the general population. However, no evidence has supported this assumption in short term effect the date [9]." This sentence is not relevant to the current study. We recommend that it be removed.

AUTHORS: Thank you for the recommendation, we fully agree and we have removed the text about covid-19.

REW:

  1. Consider whether the period of confinement by the COVID pandemic (covered by the study) has had an effect on both the modification of some of the parameters assessed.

AUTHORS: The Covid pandemic did not affect the assessment of our parameters. Our remark, which we removed from the text, concerned the increase in difficulties associated with using a PC. The original wording we used was not correct and therefore it has been modified, thank you for the good recommendation.

REW: 

  1. Please clarify the duration of the survey. Does the survey period include the COVID-19 epidemic period?

AUTHORS: The study took place in the period from July 2021 to September 2021. Data collection and analysis was not affected by the situation. Information on the duration of the study has been added to the text.

REW:

  1. Consider adding data on range of motion of the cervical joints.

AUTHORS: The range of motion in the cervical spine was measured before the actual EMG measurement in all participants. Information on the detected normal range of motion in the cervical spine was added to the text.

REW:

  1. Who performed the tests? How was the data stored ? Consider a flow diagram regarding recruitment process and data collection.

AUTHORS: The data were obtained by a specialist physiotherapist, all EMG measurements were performed under the supervision of a physician, a specialist in neurology and a specialist for EMG analysis. Information on the procedure performed and data acquisition has been added to the text.

REW:

  1. I think you are using the wrong statistical method. Since the data are from the same participants, the data must be reanalyzed to a corresponding test. The statistical analysis that should be used in this study is the Friedman test. The method of multiple comparisons is then commonly used.

AUTHORS: Thank you very much for this comment. Only you out of 3 reviewers had a proposal to change the procedure, so we would like to keep our calculations, even though we know that your proposal is also a good solution.

REW:

  1. Please clarify how the different positions affect and help in daily life, and how the different activation of the suprahyoid and infrahy-oid muscles during physiological cervical flexion may affect and help in daily life.

AUTHORS: Thank you for your recommendation. We discuss this issue in part in the discussion chapter, which has been modified and supplemented in some passages. However, in order to describe all the possibilities of how the activation of the monitored muscles can affect daily life,  we would have to extend our manuscript very much. In any case, this proposal of yours is an excellent idea for the next follow-up article.

Round 2

Reviewer 3 Report

The authors have addressed a goodly portion of comment that was raised after the initial submission. However, the authors have not addressed part comment that was raised after the submission.

It is problematic to not correct a wrong statistical method because no other reviewers have pointed out the statistical analysis.

I think it is a big problem to publish in Life with wrong statistical results.